# TAp73 Inhibits EMT and Cell Migration in Pancreatic Cancer Cells through Promoting SMAD4 Expression and SMAD4-Dependent Inhibition of ERK Activation

**DOI:** 10.3390/cancers15153791

**Published:** 2023-07-26

**Authors:** Hendrik Ungefroren, Björn Konukiewitz, Rüdiger Braun, Ulrich Friedrich Wellner, Hendrik Lehnert, Jens-Uwe Marquardt

**Affiliations:** 1Institute of Pathology, University Hospital Schleswig-Holstein, Campus Kiel, 24105 Kiel, Germany; bjoern.konukiewitz@uksh.de; 2First Department of Medicine, University Hospital Schleswig-Holstein, Campus Lübeck, 23538 Lübeck, Germany; jens.marquardt@uksh.de; 3Department of Surgery, University Hospital Schleswig-Holstein, Campus Lübeck, 23538 Lübeck, Germany; ruediger.braun@uksh.de (R.B.); ulrich.wellner@uksh.de (U.F.W.); 4University of Salzburg, 5020 Salzburg, Austria; hendrik.lehnert@plus.ac.at

**Keywords:** TAp73, PDAC, SMAD4, transforming growth factor-β, epithelial-mesenchymal transition, cell migration

## Abstract

**Simple Summary:**

Pancreatic ductal adenocarcinoma (PDAC) is a fatal disease due to early metastatic spread, late diagnosis and the lack of efficient therapies. The aim of our study was to reveal if transcriptionally active p73 (TAp73), a homolog of the well-known tumor suppressor p53, inhibits tumor progression through promoting canonical TGF-β/Smad signaling and by preventing non-canonical extracellular signal-regulated kinases (ERK)1/2-mediated TGF-β signaling. Using PDAC-derived tumor cell lines, we showed that TAp73 suppresses epithelial-mesenchymal transition by inducing the expression of epithelial markers while suppressing that of mesenchymal markers. We further demonstrated that TAp73 upregulates the expression of the TGF-β signaling intermediate SMAD4 and that SMAD4 acts a mediator of TAp73-induced inhibition of ERK activation and cell motility. Measuring the levels of TAp73 and/or SMAD4 could help to predict whether TGF-β preferentially uses an oncogenic or a tumor suppressive pathway in a given patient and at a specific time.

**Abstract:**

Pancreatic ductal adenocarcinoma (PDAC) is a fatal disease due to early metastatic spread, late diagnosis and the lack of efficient therapies. A major driver of cancer progression and hurdle to successful treatment is transforming growth factor (TGF)-β. Recent data from pancreatic cancer mouse models showed that transcriptionally active p73 (TAp73), a p53 family member, inhibits tumor progression through promoting tumor suppressive canonical TGF-β/Smad signaling, while preventing non-canonical TGF-β signaling through extracellular signal-regulated kinases (ERK)1/2. Here, we studied whether this mechanism also operates in human PDAC. Using the PDAC-derived tumor cell lines PANC-1, HPAFII and L3.6pl, we showed that TAp73 induces the expression of the epithelial marker and invasion suppressor E-cadherin and the common-mediator Smad, SMAD4, while at the same time suppressing expression of the EMT master regulator SNAIL and basal and TGF-β1-induced activation of ERK1 and ERK2. Using dominant-negative and RNA interference-based inhibition of SMAD4 function, we went on to show that inhibition of ERK activation by TAp73 is mediated through SMAD4. Intriguingly, both SMAD4 and the α isoform of TAp73—but not the β isoform—interfered with cell migration, as shown by xCELLigence technology. Our findings highlighted the role of TAp73-SMAD4 signaling in tumor suppression of human PDAC and identified direct inhibition of basal and TGF-β-stimulated pro-invasive ERK activation as an underlying mechanism.

## 1. Introduction

Pancreatic ductal adenocarcinoma (PDAC), the most common form of pancreatic cancer, has a very poor prognosis due to early metastatic spread, late diagnosis and the lack of efficient therapies [1,2,3]. This type of cancer is predicted to become the second leading cause of cancer-related deaths worldwide by 2030, which highlights the urgent need to better understand PDAC development and progression. This knowledge is crucial to identify vulnerabilities that can be targeted therapeutically to improve the patient’s fate and quality of life. Recent advances in PDAC biology focused on the intratumor microenvironment, which comprises up to 90% of the tumor mass [4,5]. The tumor stroma is mainly composed of cancer-associated fibroblasts, immune cells and excess extracellular matrix (ECM) [5], a phenomenon termed desmoplastic reaction. There is growing interest in targeting this non-malignant, but nevertheless transformed, compartment and its consequent impact on tumor development [4,6], in order to reduce tumor aggressiveness. The formation and composition of the ECM is orchestrated primarily by transforming growth factor-β (TGF-β) along with specific ECM components, the expression of which is induced by this growth factor [7]. In PDAC, TGF-β can act as both a tumor suppressor and tumor promoter depending on the cellular context and the stage of disease progression [8]. TGF-β signals primarily via the Sma and Mad-related (Smad) proteins SMAD2, 3 and 4 [9] but also through Smad independent pathways, i.e., PI3K/AKT, JNK/p38 or the extracellular signal-regulated kinase (ERK) pathway [9,10,11,12]. Alterations in the TGF-β/SMAD4 signaling pathway, particularly in *DPC4* (the gene encoding SMAD4), are critical events in PDAC progression [4].

Recently, it was revealed in two relevant and specific engineered pancreatic cancer mouse models made homozygous-null for transcriptionally active p73 (TAp73) that this p53 family member is a critical regulator of the TGF-β pathway. Mechanistically, TAp73 induces biglycan, a small proteoglycan and TGF-β inhibitor, via intermittent activation of Smad signaling [13]. The absence of TAp73 and, as a consequence, loss of expression of Smad4 and biglycan led to activation of TGF-β signaling through Smad independent pathway(s), i.e., ERK1/2, favoring oncogenic TGF-β effects and epithelial-mesenchymal transition (EMT) in the tumor cells of TAp73^−/−^ mice. The enhanced EMT phenotype of TAp73^−/−^, as compared to TAp73 wildtype (WT) cells, was associated with greater invasive abilities and a reduced sensitivity to gemcitabine treatment. The higher aggressiveness of PDAC from TAp73^−/−^ mice was most convincingly demonstrated by an increase in stromal compartment with enhanced deposition of ECM, and reduced survival of the TAp73^−/−^ mice [13]. These findings in the murine PDAC models suggest that TAp73 functions as a potent barrier to PDAC progression and implicates deletion of the *TP73* locus in PDAC initiation or progression. Although in human cancers, mutations in *TP73* are less frequent than those in *TP53*, genetic aberrations of *TP73* were, nevertheless, reported in PDAC, and previous studies already showed that loss of Tap73 induced spontaneous tumor development because of enhanced genomic instability [14]. In human pancreatic cancer cells, *TP73* monoallelic expression was also observed [15] and correlated with patient outcome [13]. With respect to functional activities, TAp73 has so far been implicated in the regulation of cell growth/death, neoangiogenesis and cellular metabolism/energy production [13,16,17].

In the PDAC mouse model carrying a loss of *TP73*, Thakur and colleagues showed that TAp73 functions as an inhibitor of the EMT process and potent barrier to PDAC progression through modulation of TGF-β signaling. In their study, the authors focused on the murine system and on TAp73-dependent regulation of the TGF-β pathway via intermittent secretion of biglycan [13]. However, the role of SMAD4 as a downstream target of TAp73 and possible upstream repressor of the ERK pathway has not been functionally dissected. Hence, it remains open whether the cellular effects of TAp73 in murine cells, particularly the induction of *DPC4* and the impact of SMAD4 on TGF-β-induced ERK activation, also operate in human PDAC. The goal of this study, therefore, was to reveal if the regulatory interactions between TAp73- and SMAD4-dependent and -independent signaling are of biological significance in human pancreatic cancer cells and whether these affect cancer relevant functions such as spontaneous and TGF-β1 dependent cell migration. Based on the data presented here, we conclude that TAp73 and SMAD4 signaling independently block basal and TGF-β1-induced ERK activation and cell migration in human PDAC-derived tumor cells.

## 2. Materials and Methods

### 2.1. Reagents

For immunoblotting, we employed the following antibodies from Cell Signaling Technology (Frankfurt am Main, Germany): anti-p73 Rabbit mAB (D3G10, #14620), anti-phospho-ERK1/2 (#4370), anti-GAPDH (14C10, #2118), and anti-Snail (#3895), as well as HRP-linked anti-rabbit (#7074) and anti-mouse (#7076) secondary antibodies; anti-HSP90 (F-8, #sc-13119) was purchased from Santa Cruz Biotechnology (Heidelberg, Germany). Recombinant human TGF-β1 (#300-023) was supplied by ReliaTech (Wolfenbüttel, Germany). An siRNA to TAp73 (5′-cgg auu cca gca ugg acg uTT-3′ and 5′-acg ucc aug cug gaa ucc gTT-3′) and a scrambled control siRNA were commercially synthesized by Metabion Int. AG (Planegg, Germany). An siRNA to SMAD4 (#1027415) was purchased from Qiagen (Hilden, Germany). The TAp73α and TAp73β vectors were a kind gift from Drs. Bertrand Joseph and Pinelopi Engskog Vlachos (Stockholm, Sweden).

### 2.2. Cells

PANC-1 human PDAC cells were obtained from the ATCC (Manassas, VA, USA), while HPAFII and L3.6pl cells were a kind gift from Dr. U. F. Wellner. Both cell lines were maintained in RPMI 1640 supplemented with 10% fetal bovine serum (FBS), 1% Penicillin-Streptomycin-Glutamine (Life Technologies, Darmstadt, Germany) and 1% sodium pyruvate (Merck Millipore, Burlington, MA, USA). PANC-1 cells stably expressing a Flag-tagged mutant version of SMAD4 (DPC4-1-514) were described and characterized in detail earlier [18,19].

### 2.3. Transfection of SiRNA and Reporter Gene Assays

On d 1, PANC-1 or HPAFII cells were seeded into Nunclon^TM^ Delta Surface plates (Nunc, Roskilde, Denmark) and transfected twice, on d 2 and 3, serum-free with 50 nM of pre-validated small interfering RNAs (siRNAs) specific for p73 or SMAD4, or scrambled siRNAs as control for 4 h, using Lipofectamine 2000 (PANC-1) or RNAiMAX (HPAFII) (both from Life Technologies/Thermo Fisher Scientific, Waltham, MA, USA) according to the manufacturer‘s instructions. Transfected cells were subjected to immunoblot analyses, reporter gene assays, or cell migration assays. For reporter gene assays, PANC-1 cells were seeded in 96-well plates on d 1 and co-transfected on d 2 serum-free with various siRNAs and Lipofectamine 2000. Twenty-four h later (d 3), cells again received Lipofectamine 2000 with the same siRNAs together with p(CAGA)_12_ MLP-Luc and the *Renilla* luciferase encoding vector pRL-TK-Luc. On d 4, cells were treated with TGF-β1 for 24 h and luciferase activities were determined with the Dual Luciferase Assay System (Promega, Mannheim, Germany). In all reporter gene assays, the data were derived from 6 wells processed in parallel and normalized with *Renilla* luciferase activity as described in detail elsewhere [7].

### 2.4. RT-PCR Analysis

Following transfection of PANC-1, HPAFII or L3.6pl cells, total RNA was purified by affinity chromatography (Qiagen) and reverse transcribed as described in detail earlier [7,18,20]. Relative mRNA expression of target genes was measured with quantitative real-time PCR and Maxima SYBR Green Mastermix (Thermo Fisher Scientific) using an I-Cycler (BioRad, Munich, Germany). Expression of the genes of interest was normalized to that of glyceraldehyde-3-phosphate dehydrogenase (GAPDH). All PCR primer sequences were supplied in previous publications [7,18,20].

### 2.5. Immunoblotting

The procedure for immunoblotting was described in detail earlier [7,18]. Briefly, cells were lysed with 1 × PhosphoSafe lysis buffer (Merck Millipore). Equal amounts of proteins were fractionated by SDS-PAGE on mini-PROTEAN TGX any-kD precast gels (BioRad) and blotted to PVDF membranes. After blockage with nonfat dry milk or bovine serum albumin, membranes were incubated with primary antibodies overnight at 4 °C. After incubation with HRP-linked secondary antibodies, chemiluminescent detection of proteins was performed on a ChemiDoc XRS imaging system (BioRad) using Amersham ECL Prime Detection Reagent (GE Healthcare, Hong Kong, China). Quantification of band intensities for the proteins of interest was carried out by densitometric readings and normalization to those for GAPDH or HSP90 in the same sample.

### 2.6. Real-Time Cell Migration and Invasion Assays

We employed the xCELLigence^®^ DP system (ACEA Biosciences, San Diego, CA, USA) to measure random/spontaneous cell migration in a chemokinesis setup, according to previous descriptions [18,20]. To enhance the adhesion of cells to the lower side of transwell membrane equipped with the gold electrodes, it was coated with a 1:1 mixture of collagens I and IV. Following assembly of the CIM plate-16 and a 1 h equilibration in an incubator, each well of the CIM plates received 60,000–80,000 cells in standard growth medium supplemented with 1% (rather than 10%) FBS to minimize proliferation. Data acquisition was carried out at intervals of 15 min and the assays were run for various lengths of time and analyzed with RTCA software (version 1.2, ACEA Biosciences). The setup for measuring invasion was identical, except that the surface of the upper chamber was covered with a monolayer of 5% (*v*/*v*) growth factor-reduced Matrigel (BD Biosciences, Heidelberg, Germany), diluted 1:20 with RPMI 1640 basal medium.

### 2.7. Statistical Analysis

Statistical significance was calculated using the unpaired two-tailed Student t test or the Wilcoxon test. Results were considered significant at *p* < 0.05 and denoted in the graphs by an asterisk or a rhombus.

## 3. Results

### 3.1. TAp73 Up-Regulates ECAD and SMAD4 in in Human PDAC Cells

Thakur and colleagues revealed in murine cells that TAp73 inhibited EMT by inducing the expression of *CDH1* and other epithelial genes, while suppressing that of mesenchymal genes as well as EMT-associated functions like cell migration [13]. We, therefore, set out to study the impact of TAp73 on EMT marker expression in human cells employing primarily the human PDAC cell lines, PANC-1, HPAFII and L3.6pl, a metastatic variant of the SMAD4-positive cell line COLO 357.

We first transfected PANC-1, HPAFII or L3.6pl cells with a p73-specific siRNA and monitored the transfectants by immunoblotting for successful downregulation of TAp73α protein levels (Appendix A). We detected only one band of low intensity that displayed the same electrophoretic mobility as TAp73α ectopically expressed in PANC-1 cells and run side-by-side as a control (Appendix A). The absence of a second band of higher mobility strongly suggested that TAp73α is the only—or at least the most abundant—p73 isoform expressed in these cells. The low endogenous protein levels of TAp73α in all three PDAC lines might be due to low stability of the WT TAp73 protein similar to WT p53 protein. Indeed, an increase in the half-life of the p73 protein and its subsequent accumulation was observed in HCT116 cells upon treatment with cisplatin [21]. Next, we analyzed the p73 siRNA-transfected PANC-1 cells by qPCR analysis for expression of epithelial marker genes. Interestingly, knockdown of p73 reduced the mRNA abundance of ECAD, Grainyhead-like 2 (GRHL2), an epithelial-specific transcription factor, and SMAD4 (Figure 1A). Regulation of ECAD and SMAD4 by p73 was also evident at the protein level (Figure 1B). Similar results with respect to ECAD and SMAD4 regulation by TAp73 were obtained for HPAFII and L3.6pl cells (Appendix A).

The p73 siRNA employed here likely inhibited more than one isoform derived from *TP73*, i.e., not only TAp73α but also TAp73β. To obtain information as to whether both isoforms differ in their ability to stimulate expression of *CDH1*, *GRHL2* and *DPC4*, we transfected PANC-1 cells with expression vectors for TAp73α or TAp73β and initially monitored the transfectants by immunoblotting for successful ectopic expression of the transgenes. Both isoforms were expressed with TAp73β exhibiting a higher electrophoretic mobility than TAp73 α due to lack of the SAM domain [22] (Figure 1C). In qPCR analysis of the transfectants, we found increased mRNA levels for ECAD, GRHL2 and SMAD4 in response to transfection of TAp73α, while that of TAp73β did not exhibit a statistically significant effect on either gene (Figure 1D). We, therefore, focused here on TAp73α, since expression of this isoform is the major outcome of *TP73* gene expression and the relevance and contribution of the other isoforms remain unclear [22].

### 3.2. Knockdown of TAp73 Interfered with TGF-β1-Induced Luciferase Activity on a SMAD-Responsive Promoter and Regulation of TGF-β/SMAD Target Genes in Human PDAC Cells

Above, we showed that TAp73 sustains the protein levels of SMAD4, suggesting that TAp73 may impact on Smad-mediated transcriptional activation of TGF-β1 target genes. To demonstrate this more directly, PANC-1 cells silenced for TAp73 were transiently transfected with a TGF-β/Smad-responsive reporter gene and monitored for their sensitivity to TGF-β1 stimulation. We employed the p(CAGA)_12_ MLP-luc plasmid rather than p3TPlux used by Thakur and colleagues [13]. Unlike p3TPlux, which is also sensitive to TGF-β dependent non-Smad-mediated transcriptional activation [21], p(CAGA)_12_ MLP-luc carries twelve Smad binding elements (SBEs) in tandem and, hence, only responds to SBE binding of SMAD3/4 [23]. With this reporter, TGF-β1-induced firefly luciferase activity was dramatically reduced in cells that received either p73 siRNA, or SMAD4 siRNA as positive control, compared to scrambled siRNA-transfected cells (Figure 2A). This suggests that in pancreatic tumor cells of human origin, too, TAp73 promotes TGF-β signaling through activation of a SMAD4-dependent pathway.

The failure of TGF-β1 to induce luciferase activity in a TAp73-deficient context suggested that TAp73 promotes TGF-β signaling through activation of Smad signaling. If so, this should also impact the response of TGF-β target genes in human cells that are regulated in a SMAD-dependent manner such as *CDH1*, *TGFB1* and *SNAI1*. To this end, when compared to control siRNA transfected cells, in which the inhibitory effect of TGF-β1 treatment on *CDH1* was 81.3% after 24 h of treatment, this was reduced to 21.5% in TAp73-silenced PANC-1 cells. However, in the absence of TGF-β1, basal levels of ECAD in p73 siRNA transfected cells dropped to 17.5% of control siRNA transfected cells (Figure 2B). This is consistent with the results shown in Figure 1A and, in addition, suggests that TAp73 is required for TGF-β1 to efficiently down-regulate ECAD expression. The stimulatory effect of TGF-β1 on *TGFB1* (a gene partially dependent on SMADs and ERK) dropped from 27-fold to 10-fold (Figure 2B). Moreover, basal protein levels of ECAD or SNAIL, a transcriptional regulator of *CDH1*, were decreased or increased, respectively, in TAp73-silenced PANC-1 cells (consistent with the anti-EMT function of TAp73 and previously observed also in TAp73-deficient PDAC cells from mice [13]). However, down- or upregulation of the ECAD or SNAIL proteins, respectively, by exogenous TGF-β1 in the TAp73-silenced cells was alleviated (Figure 2C, lanes 3 and 4). We conclude from these data that in PDAC cells, TAp73 is required for a full-blown response of SMAD-dependent TGF-β target genes to TGF-β1 treatment, most likely by its ability to promote SMAD4 expression.

### 3.3. TAp73 Inhibits Basal and TGF-β1-Induced ERK Activation

Thakur and coworkers showed that TAp73 knockout in mice caused downregulation of Smad4 in vivo, as demonstrated by immunohistochemistry in PDAC tissues and in cell lines derived from TAp73-deficient mice compared with control mice [13], and that non-Smad, i.e., ERK signaling, is enhanced [13]. We, therefore, asked whether in human PDAC cells, too, TAp73 suppresses ERK activation. Knockdown of p73 in PANC-1 cells by siRNA enhanced both the basal (Figure 3A, left-hand graph) and TGF-β1-induced levels (Figure 3A, right-hand graph) of phospho-ERK1/2 (pERK1/2). The inducive effect of TGF-β1 was maximal at 0.25 h and subsequently declined when monitored over a period of 2 h, and was generally greater for ERK2 than for ERK1. Very similar effects were observed in HPAFII cells (Figure 3B). Our results, therefore, suggest that ERK activation is suppressed by TAp73 in pancreatic cancer cells.

### 3.4. The Inhibitory Effect of TAp73 on ERK Activation Is Mediated via SMAD4

Given the induction of SMAD4 by TAp73 (Figure 1B), the question arose as to whether TAp73-mediated inhibition of ERK activation is mediated by SMAD4, or in other words, whether SMAD4 can mimic the effects of TAp73 on ERK activation. To explore this in more detail, we employed previously characterized PANC-1 cells, in which WT SMAD4 function is inhibited by a stable ectopic expression of C-terminally truncated SMAD4 protein, SMAD4-(1-514), which acts in a dominant-negative fashion [18,19]. When these cells were monitored for the activation status of ERK1/2, we noted an increase in the levels of pERK1/2 under basal conditions (Figure 4A) and after challenge with exogenous TGF-β1, peaking at 0.25 h of TGF-β1 stimulation (Figure 4B). To confirm this observation, we knocked down SMAD4 in PANC-1 (Figure 4C) and HPAFII (Figure 4D) cells by siRNA followed by ERK phospho-immunoblotting. The results obtained were consistent with those of the dominant-negative inhibition approach, namely that pERK1/2 activation was enhanced in p73 knockdown cells with the greatest abundance of pERK1 and 2 seen at the 0.25 h time point. Taken together, these data show that SMAD4 inhibition phenocopied the stimulatory effect of the p73 knockdown on ERK1/2 activation.

### 3.5. TAp73α and SMAD4 Mediate Inhibition of Cell Migration in Human PDAC Cells

Thakur and colleagues [13] observed that TAp73-deficient murine tumor cells harbor an EMT phenotype associated with enhanced invasive properties. Above, we showed that TAp73 promoted expression of ECAD, an epithelial marker and a well-known invasion inhibitor, but inhibited that of SNAIL, a mesenchymal marker and driver of invasion. To reveal whether enhanced migration or invasion is also a consequence of silencing TAp73 in human PDAC cells, we transfected PANC-1 or HPAFII cells with the TAp73-specific siRNA and subjected them to real-time measurement of random cell migration on an xCELLigence platform. Of note, knockdown of TAp73 in PANC-1 cells resulted in a moderate promigratory effect under basal conditions (Figure 5A, left-hand graph) and a more pronounced one under TGF-β1 stimulation (Figure 5A, right-hand graph). A similar migration profile was seen with HPAFII cells silenced for p73 (Appendix A) and with PANC-1-p73 knockdown cells in an invasion setup of the RTCA assay using Matrigel as barrier (Appendix A). Conversely, ectopic overexpression of TAp73α (verified by p73 immunoblotting, Figure 1C) inhibited both basal and TGF-β1-induced migration in PANC-1 cells (Figure 5B, left-hand graph), while that of TAp73β (lacking the SAM domain) had the opposite effect (Figure 5B, right-hand graph). We, therefore, conclude that TAp73α, but not TAp73β, inhibits cell migration in human PDAC-derived tumor cells.

Above, we showed that TAp73 induced SMAD4 and that both TAp73 and SMAD4 inhibited basal and TGF-β1-induced ERK activation. This suggested the possibility that migration inhibition by TAp73, too, is mediated through SMAD4. To test this directly, we again utilized PANC-1 cells stably expressing dominant-negative SMAD4-(1-514) and subjected these cells to xCELLigence migration assay. Intriguingly, impairing WT SMAD4 function enhanced both basal and TGF-β1-induced migratory activity in two individual clones (Figure 5C), and thus mimicked the effect of the TAp73 knockdown on cell migration.

## 4. Discussion

Despite recent advances in chemotherapeutic treatments, the prognosis for PDAC is still poor and urgently requires a deeper understanding of the molecular events and critical signaling pathways that drive tumor development and evolution. Previous studies reported that the p53 homolog TAp73 is involved in cancer development through cell growth and death regulatory mechanisms. However, the significance of its altered expression in various cancers, including PDAC, has not yet been clearly defined. Using endogenous mouse models of PDAC, a pioneering study by Thakur and coworkers investigated the role of TAp73 in pancreatic carcinogenesis and showed that TAp73 deficient PDAC exhibited characteristics of EMT and enhanced desmoplasia, suggesting enhanced activity of TGF-β [13]. Interestingly, the increased amount of free TGF-β, which is suspected to be associated with a higher risk of pancreatic cancer [24] and resistance to anticancer treatment [25], resulted from the inability to trigger activation of the Smad dependent pathway (primarily due to downregulation of Smad4) and to induce expression of the Smad4 target and TGF-β inhibitor biglycan. As a consequence of the increased levels of free TGF-β, the tumor cells can display high levels of Smad independent pathway activation, such as that involving ERK derepressed under conditions of TAp73 deficiency [13]. These favor the expression of EMT-related transcription factors such as Snail and Zeb and promote EMT, enhanced migratory capacity and invasiveness as well as resistance to chemotherapeutic agents in TAp73-deficient murine PDAC cells. The data from murine cells suggest that both cell-intrinsic and paracrine effects brought about by TAp73 deficiency force a switch in function of TGF-β in carcinogenesis from tumor suppressive to tumor promoting [13].

While plentiful data from mice were presented in the Thakur study, any conclusions as to whether this TAp73-driven network also operates in human PDAC remained open. A clue that TAp73 is involved in EMT regulation in human cells, however, came from the observation that knockdown of TAp73 induced MCF10A mammary epithelial cells to undergo EMT via downregulation of *CDH1* and upregulation of *SNAI1*, and an increase in cell migration [26]. By suppressing EMT, TAp73 physiologically achieves maintenance of normal cell polarity in these cells [26]. This is consistent with the role of TAp73 in cellular differentiation and suppression of the mesenchymal phenotype [27,28,29].

In our study, we initially found evidence for a tumor suppressive role of TAp73 in PDAC cells of human origin by observing that TAp73 upregulated the basal expression of ECAD and SMAD4, while downregulating that of SNAIL. By employing luciferase assays with a strictly SMAD-responsive reporter plasmid, we were, in addition, able to reveal that TAp73 promotes TGF-β signaling through activation of a SMAD4-dependent pathway. In line with this, TAp73 was required for TGF-β1 regulatory effects on TGF-β/SMAD-dependent target genes, like *CDH1* (downregulation) and *TGFB1* (upregulation). In addition to its role as a promoter of epithelial gene expression, TAp73 was identified in murine cells as an inhibitor of mesenchymal gene expression, i.e., Snail, and mesenchymal non-Smad signaling, i.e., Erk1/2 [13], two functions that we demonstrated here to operate also in human PDAC cells. The coordinated induction of epithelial genes and concurrent suppression of mesenchymal genes and pathways suggests that TAp73 is also a crucial antagonist of EMT and cell motility in human PDAC cells.

Regarding Smad independent signaling, we observed in both PANC-1 and HPAFII cells constitutive ERK pathway activation that was enhanced as a result of TAp73 silencing. Since we previously showed that exogenous TGF-β1 was able to rapidly stimulate ERK1/2 activation in PANC-1 and other PDAC cell lines that retained sensitivity to this growth factor [18], we asked whether another tumor suppressive function of TAp73 could be repression of TGF-β1-driven ERK activation. To this end, in TAp73-depleted cells, short-term ERK activation was significantly enhanced. Moreover, using dominant-negative and RNA interference with WT SMAD4 function in two different PDAC lines, we were able to show that higher ERK activation was a direct consequence of SMAD4 inhibition.

Prompted by suppression of the EMT phenotype, we reasoned that TAp73 should also interfere with cell migration/invasion. Thakur and colleagues already showed that RNAi-mediated knockdown of TAp73 in PANC-1 cells enhanced (directed) migratory activity using a Boyden chamber assay with a chemotaxis setup and FBS as attractant [13]. Here, we applied the real-time xCELLigence technology in a chemokinesis setup to monitor the TGF-β1-dependent migratory potential of PANC-1 and HPAFII cells after knockdown of TAp73 or ectopic expression of either TAp73α or TAp73β. These assays showed strong de-repression of both spontaneous and TGF-β1-driven migratory and invasive activities in the p73 knockdown cells its inhibition after ectopic expression of TAp73α but not TAp73β. Finally, to verify that the anti-migratory effect of TAp73 was due to the induction of SMAD4 expression, we performed migration assays with PANC-1 cells harboring defective SMAD4 function as a result of ectopic expression of dominant-negatively acting SMAD4 mutant. In accordance with the p73 siRNA data, cells with defective SMAD4 function also exhibited an increased migratory activity. This is consistent with studies in other PDAC-derived tumor cell lines, in which SMAD4 acted as an inhibitor of migration or invasion [30,31] confirming the tumor suppressing function of SMAD4 in PDAC cells. The observation that both TAp73 and SMAD4 target ERK1/2 activation for inhibition are in perfect agreement with earlier findings from us [32] and others [12] on the crucial role of this pathway in driving EMT and migration/invasion in PDAC. Moreover, suppression of ERK activation by (endothelial) Smad4 was demonstrated to restrain the transition to hematopoietic progenitors [33]. Our findings on the role of TAp73 in negative regulation of TGF-β dependent EMT and cell migration in human PDAC cells have been summarized in Figure 6.

While the primary goal of this study was to demonstrate that the original findings on TAp73 in murine PDAC cells also operate in their human orthologues, we provided additional data that were not contained in the Thakur study. Specifically, we revealed that only the α, but not the β, isoform was able to stimulate ECAD and SMAD4 expression (Figure 1), and to inhibit spontaneous and TGF-β1-driven cell migration and invasion, while the latter response was even promoted by the β isoform (Figure 5). TAp73β differed from TAp73α by the lack of the SAM domain, a potential protein–protein interaction domain that might contribute to the control of TAp73 transcriptional activity [22]. Mechanistically, TAp73 may act via transactivation of the *DPC4* gene promoter, which harbors a p53 response element [13].

In the present study, we provided evidence that TAp73 through induction of *DPC4* suppresses basal and TGF-β1 dependent activity of the MEK-ERK signaling pathway. According to the model proposed by Thakur et al., de-repression of ERK activation and an ensuing increase in EMT and migration/invasion—rather than being executed directly by SMAD4—is believed to be an indirect effect, ultimately resulting from increased levels of free TGF-β due to the absence of TGF-β trapping by biglycan that, in turn, is induced via TAp73 and SMAD4 [7,13]. The issue of whether in human PDAC, too, biglycan and endogenous TGF-β are involved here—although being highly relevant—was beyond the scope of the present study.

Altogether, our data on human PDAC clearly suggest that the absence of TAp73 impairs TGF-β signaling toward the tumor-suppressing SMAD4 dependent pathway. Hence, TAp73 in suppressing EMT and cell motility might have implications for other tumor-suppressive functions, e.g., responsiveness to SMAD4-dependent cell death after TGF-β treatment [34]. It will be interesting to see if TAp73 deficiency can render PANC-1, HPAFII or L3.6pl cells less apoptosis-sensitive to TGF-β/SMAD4-dependent cell death. Moreover, we are currently planning murine xenotransplantation experiments to study if after intra-pancreatic injection of human TAp73-deficient cells (as observed for the murine counterparts [13]), the number of liver metastases that developed from these cells is higher than that with the TAp73 WT cells. These experiments might also reveal if the loss of both tumor-suppressive functions (anti-EMT/anti-invasion and apoptosis) contributes to the pro-metastatic effect.

## 5. Conclusions

Our findings, which highlighted the complex role of TGF-β in pancreatic tumorigenesis, might have implications for therapeutic approaches targeting this growth factor for inhibition. Currently, several inhibitors of TGF-β receptors or ligands are being evaluated in clinical trials. For instance, galunisertib, the first small molecule TGF-β receptor inhibitor, plus gemcitabine, resulted in the improvement of survival in patients with unresectable PDAC [35]. However, the lack of predictive biomarkers to identify patients likely to respond makes PDAC treatment with TGF-β inhibitors a challenging issue. Careful patient selection and timing of treatment with respect to activation of TGF-β signaling could help to improve this situation. Hence, measuring the levels of TAp73, SMAD4 and/or pERK could help to predict whether TGF-β preferentially uses an oncogenic or a tumor-suppressive pathway in a given patient and at a specific time.

## Figures and Tables

**Figure 1 cancers-15-03791-f001:**
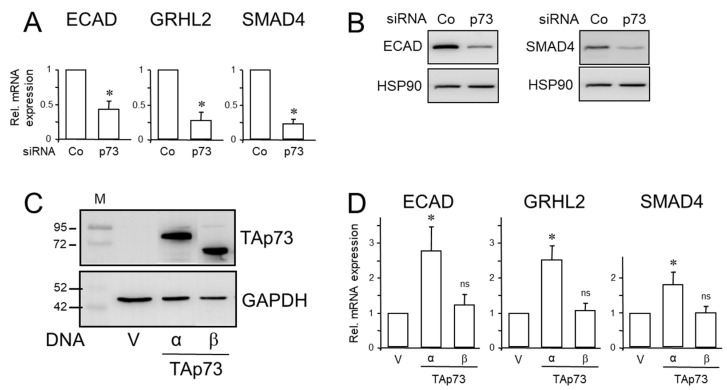
TAp73 induces the expression of epithelial genes and SMAD4. (**A**) PANC-1 cells were transiently transfected twice with 50 nM each of p73 siRNA or an irrelevant control (Co) siRNA and 48 h after transfection subjected to qPCR for ECAD, GRHL2 and SMAD4, as well as GAPDH as normalization control. Data represent the normalized mean ± SD of three assays. (**B**) As in (**A**), except that cells were subjected to immunoblotting for ECAD (left-hand panel) or SMAD4 (right-hand panel), and HSP90 as loading control. Successful knockdown of p73 was verified by immunoblot analysis (see Appendix A). (**C**) Immunoblot analysis show successful overexpression of TAp73α and TAp73β in PANC-1 cells transfected with either empty vector (V) or expression vectors for the two p73 isoforms and incubated with a p73 antibody. The bands of the endogenous p73 protein are not visible here due to the short exposure time of the blot. (**D**) PANC-1 cells were transiently transfected with empty vector (V) or expression vectors for TAp73α or TAp73β, and processed for qPCR analysis of ECAD, GRHL2 and SMAD4, and GAPDH as a control. Data represent the normalized mean ± SD of three experiments. The asterisks (*) in (**A**,**D**) denote a significant difference relative to the control (*p* < 0.05, Wilcoxon test); ns, non-significant.

**Figure 2 cancers-15-03791-f002:**
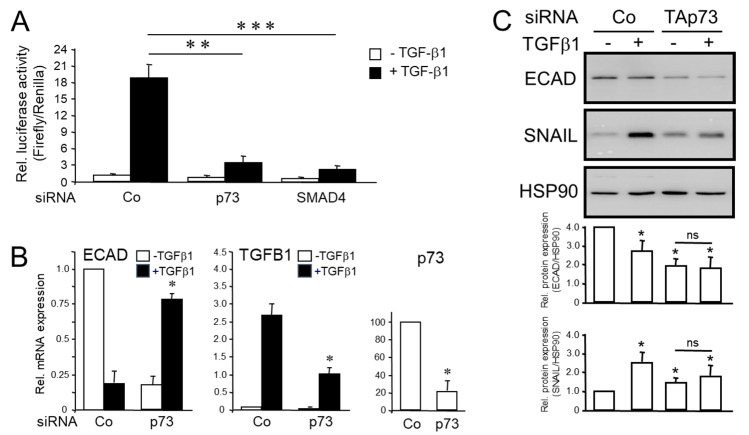
Knockdown of TAp73 interfered with TGF-β/Smad-specific transcriptional activation and TGF-β1-induced regulation of TGF-β target genes in human pancreatic tumor cells. (**A**) PANC-1 cells (10,000) were seeded in 96 wells on d 1 and were transfected on d 2 with RNAiMAX along with 50 nM each of negative (scrambled) control (Co) siRNA, p73 siRNA or SMAD4 siRNA. On d 3, cells received the same siRNAs along with 20 ng/well of p(CAGA)_12_ MLP-luc, and 5 ng/well of the *Renilla* luciferase encoding vector pRL-TK-Luc using Lipofectamine 2000. Forty-eight h after the start of the first transfection, cells were stimulated with TGF-β1 (5 ng/mL) for another 24 h. At the end of the stimulation period, cells were lysed in Glo lysis buffer and subjected to luciferase assay. Relative firefly luciferase activities were measured in PANC-1 cell extracts, normalized to those for *Renilla* luciferase. Three independent experiments were performed with similar results. Data shown are the mean ± SD of six wells processed in parallel. The asterisks (*) indicate significance vs. TGF-β1-treated Co cells (unpaired two-tailed Student’s *t*-test; ** *p* < 0.01, *** *p* < 0.001). (**B**) PANC-1 cells were treated for 24 h with recombinant human TGF-β1 and subjected to qPCR for ECAD or TGFB1 and p73 (from left to right) to verify successful knockdown. Data are the mean ± SD of three experiments (*p* < 0.05, Wilcoxon test). (**C**) As in (**B**), except that cells were subjected to immunoblot analysis of ECAD, SNAIL and HSP90 as a loading control. The graphs underneath the blots show densitometry-based signal quantification from three independent blots (mean ± SD, *n* = 3). The asterisks (*) indicate significant differences relative to non-TGF-β1 treated controls (*p* < 0.05, Wilcoxon test); ns, non-significant.

**Figure 3 cancers-15-03791-f003:**
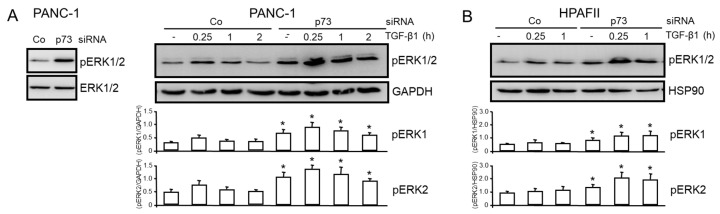
TAp73 interferes with basal and TGF-β1-induced ERK activation. (**A**) PANC-1 cells were transiently transfected with 50 nM each of p73 siRNA or an irrelevant control (Co) siRNA and were either left untreated (left-hand blot) or were treated 48 h after transfection for various times (as indicated) with 5 ng/mL TGF-β1. Subsequently, cells were subjected to immunoblotting of pERK1/2 and either total ERK1/2 (left-hand blot) or GAPDH (right-hand blot) to control for equal loading. Since we observed that within the 2 h observation period, neither GAPDH nor HSP90 expression was affected by modulation of TAp73 expression, these housekeeping proteins rather than total ERK1/2 were used for normalization of pERK1 and pERK2 levels. (**B**) The same as in (**A**), except that HPAFII cells were used. Successful knockdown of TAp73 in (**A**) was verified by immunoblotting (Appendix A) and in (**B**) by qPCR. The graphs below the blots show results from densitometry-based signal quantification of three experiments (means ± SD, *n* = 3). The asterisks (*) indicate significance (*p* < 0.05, Wilcoxon test) relative to the respective Co siRNA sample.

**Figure 4 cancers-15-03791-f004:**
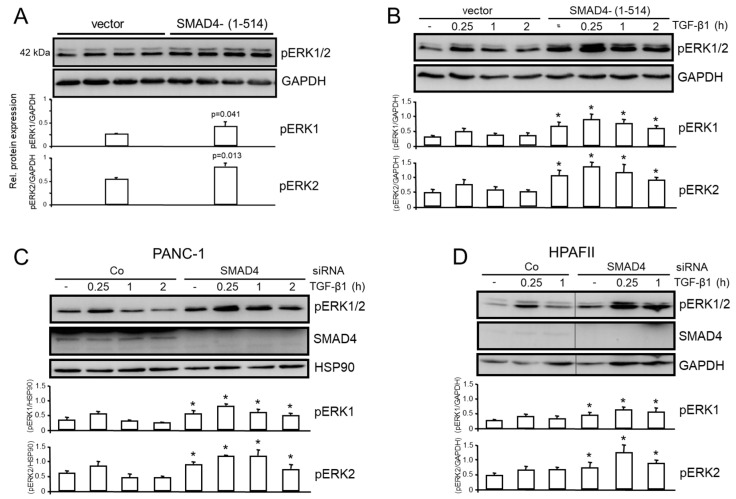
PANC-1 cells stably expressing (**A**) a C-terminally truncated form of SMAD4, SMAD4-(1-514) or empty vector controls (quadruplicate samples each) were either left untreated, or (**B**) were treated with TGF-β1 for the indicated times, and subjected to pERK1/2 immunoblotting. The data shown in (**A**) are representative of three experiments. The graphs in (**A**) represent the densitometric values (means ± SD) of four wells after normalization to those for GAPDH. The *p* values were calculated with the unpaired two-tailed Student *t*-test. (**C**) PANC-1 cells were transiently transfected with siRNAs to SMAD4 followed by immunoblotting of pERK1/2, and HSP90 to control for equal loading. (**D**) As in (**C**), except that HPAFII cells were employed and GAPDH as a loading control. The quantitative data in (**B**–**D**) (graphs underneath the blots) represent the normalized densitometric data derived from three independent experiments (means ± SD, *p* < 0.05, Wilcoxon test). The asterisks (*) indicate significant differences relative to the respective Co siRNA transfected cells.

**Figure 5 cancers-15-03791-f005:**
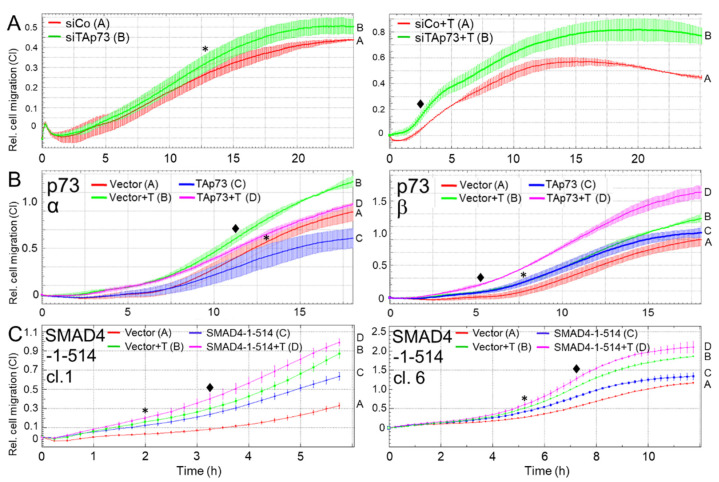
TAp73 and SMAD4 inhibit cell migration in human PDAC-derived tumor cells as revealed by xCELLigence-based real-time assay. (**A**) PANC-1 cells were transiently transfected twice (on two consecutive days) with 50 nM of either control siRNA (siCo) or TAp73 siRNA (siTAp73) and subsequently subjected to cell migration assay with xCELLigence technology in the absence (left-hand graph) or presence (right-hand graph) of exogenous TGF-β1 (+T, 5 ng/mL). Measurements of migratory activity were taken every 15 min and graphically displayed as the dimensionless cell index (CI) plotted against assay time in h. Data are from a representative experiment out of three experiments performed in total (mean ± SD from quadruplicate wells). Successful knockdown of TAp73 was verified by immunoblotting and qPCR analysis (see Figure 2B and Appendix A). (**B**) PANC-1 cells were transfected with either TAp73α (p73α, left-hand graph) or TAp73β (p73β, right-hand graph), or empty vector as control, and 48 h later assayed as in (**A**) for migratory activity in the absence or presence (+T) of TGF-β1. In each panel, the assay shown is representative of three assays performed in total. Data are the means ± SD from 3–4 parallel wells. Successful overexpression of the TAp73α and TAp73β transgenes is shown in Figure 1C. (**C**) Two individual clones (cl. 1 and 6) of PANC-1-SMAD4-(1-514) cells were subjected to cell migration assay in the absence or presence of rec. human TGF-β1 (+T, 5 ng/mL). For each clone, a representative experiment is shown out of three experiments performed in total (mean ± SD from 3–4 parallel wells). The asterisks (*) indicate the first time point at which differences are significantly different between the untreated control cells (siRNA in panel **A**, vector in panels **B**,**C**) and the cells transfected with a TAp73-specific nucleic acid (siRNA in panel **A** and TAp73 encoding vector in panels **B**,**C**). The rhombuses (♦) indicate the first time point at which differences are significantly different between the corresponding TGF-β1-treated cell populations (*p* < 0.05, unpaired two-tailed Student’s *t*-test).

**Figure 6 cancers-15-03791-f006:**
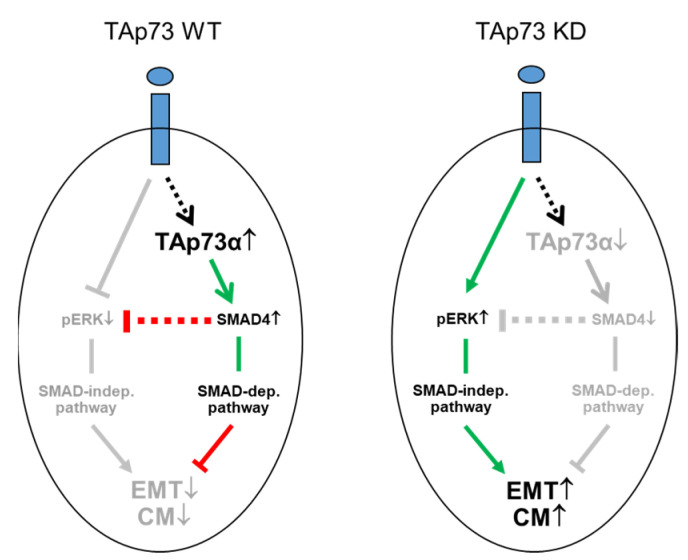
Cartoon illustrating the role of TAp73 in negative regulation of TGF-β dependent EMT and cell migration (CM) in human PDAC cells. Left-hand side, in WT TAp73 epithelial tumor cells, TAp73α induces SMAD4 expression and following stimulation of cells with TGF-β1 (blue ovals) and activation of its receptors (blue rectangles), SMAD4 inhibits the basal and TGF-β1-induced formation of pERK and an ERK-mediated increase in EMT and CM. Right-hand side, following siRNA-mediated knockdown (KD) of TAp73α, the subsequent decrease in SMAD4 expression and a failure to activate a SMAD-dependent (dep.) pathway, reinforced TGF-β signaling switching to SMAD4-independent (indep.) pathways, e.g., MEK-ERK1/2, by removing the inhibitory effect of SMAD4 on pERK1/2 formation. In the course of this study, we also carved out that only the α isoform of TAp73 (TAp73α) is able to promote SMAD4 and ECAD expression and to inhibit CM. The green arrows indicate activation, while the red lines indicate suppression. Grey-shaded arrows and lines indicate the inactive state. The stippled lines and arrows denote the possibility that these effects are indirect. For details see text.

## Data Availability

All data reported are contained in the Section 3 and the Appendix A.

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
