# Peer review of "TAp73 Inhibits EMT and Cell Migration in Pancreatic Cancer Cells through Promoting SMAD4 Expression and SMAD4-Dependent Inhibition of ERK Activation"

_cancers, 2023, doi:10.3390/cancers15153791_

Round 1
Reviewer 1 Report
In this article, the tumor suppressor protein TAp73 and transforming growth factor (TGF)- are discussed in relation to pancreatic ductal adenocarcinoma (PDAC). The study examines whether the tumor-suppressive effects of TAp73 previously reported in mice models through TGF-/Smad signaling also function in human PDAC. The researchers show that TAp73 promotes the expression of E-cadherin and SMAD4, which are linked to decreased invasion and tumor suppression. This is done using PDAC-derived cell lines. TAp73 simultaneously inhibits ERK1/2 activation and reduces SNAIL expression. The research also demonstrates that TAp73's suppression of ERK activation is brought about via SMAD4. It's interesting to note that SMAD4 and the TAp73 isoform both prevent cell migration, highlighting their functions in preventing tumor growth. Overall, our results shed insight on the mechanism of ERK inhibition and emphasize the significance of TAp73-SMAD4 signaling in preventing tumor development in human PDAC.
The following experiment are recommended to enhance the quality of the study.
1. Examining the impact of TAp73 knockdown on the expression of EMT markers in additional PDAC cell lines.
2. To expand the study's scope, choose other human PDAC cell lines (other than PANC-1 and HPAFII) for the experiment.
3.Use a p73-specific siRNA to carry out TAp73 knockdown in the chosen PDAC cell lines, using the same steps as with PANC-1 cells.
3. Use immunoblotting to validate that the transfected cells' TAp73 protein levels were successfully downregulated, as shown in Figure S1.
4. Use qPCR and immunoblotting, respectively, to examine the mRNA and protein levels of the EMT marker genes (ECAD, GRHL2, and SMAD4) in the TAp73 knockdown cells. Compared to the control cells, examine the findings.
5. For the experiment to be statistically significant and the results to be reliable, do it more than once using replicates.
6. To account for any possible off-target effects, use positive controls, such as siRNA-transfected cells that have been scrambled.
7. Provide a thorough explanation of the data and conduct out in vivo experiments to comprehend TAp73's function in PDAC's EMT regulation.
8- The Authors also need to perform the invasion assay.
Quality of English Language is fine
Author Response
Dear Editor:
We thank the reviewers for their critical and generally very positive comments on our manuscript. However, we only disagree with some notions of Reviewer 2. We believe that incorporating the suggestions and recommendations of all three reviewers into the revised version has significantly enhanced the quality of our manuscript.
Reviewer 1
In this article, the tumor suppressor protein TAp73 and transforming growth factor (TGF)- are discussed in relation to pancreatic ductal adenocarcinoma (PDAC). The study examines whether the tumor-suppressive effects of TAp73 previously reported in mice models through TGF-/Smad signaling also function in human PDAC. The researchers show that TAp73 promotes the expression of E-cadherin and SMAD4, which are linked to decreased invasion and tumor suppression. This is done using PDAC-derived cell lines. TAp73 simultaneously inhibits ERK1/2 activation and reduces SNAIL expression. The research also demonstrates that TAp73's suppression of ERK activation is brought about via SMAD4. It's interesting to note that SMAD4 and the TAp73 isoform both prevent cell migration, highlighting their functions in preventing tumor growth. Overall, our results shed insight on the mechanism of ERK inhibition and emphasize the significance of TAp73-SMAD4 signaling in preventing tumor development in human PDAC.
The following experiment are recommended to enhance the quality of the study.
- Examining the impact of TAp73 knockdown on the expression of EMT markers in additional PDAC cell lines.
Response: We have added corresponding expression data from a third PDAC cell line, namely L3.6pl, a metastatic variant of the PDAC cell line COLO 357 (PMID: 10935470).
- To expand the study's scope, choose other human PDAC cell lines (other than PANC-1 and HPAFII) for the experiment.
Response: As outlined under point 1, we have employed L3.6pl cells.
3. Use a p73-specific siRNA to carry out TAp73 knockdown in the chosen PDAC cell lines, using the same steps as with PANC-1 cells. Use immunoblotting to validate that the transfected cells' TAp73 protein levels were successfully downregulated, as shown in Figure S1.
Response: We have used the same p73-specific siRNA as in PANC-1 cells to knock down TAp73 also in HPAFII and L3.6pl cells. The resulting protein lysates were fractionated on the same gel as the p73 siRNA-transfected PANC-1 samples and immunoblotted for TAp73. These data have been added to Figure S1 as panel B.
- Use qPCR and immunoblotting, respectively, to examine the mRNA and protein levels of the EMT marker genes (ECAD, GRHL2, and SMAD4) in the TAp73 knockdown cells. Compared to the control cells, examine the findings.
Response: This suggestion is related to that in point 1. We have done this for L3.6pl cells.
- For the experiment to be statistically significant and the results to be reliable, do it more than once using replicates.
Response: As indicated in the figure legends, we have performed most experiments at least three times and have calculated the means, standard deviations, and p values from n = 3.
- To account for any possible off-target effects, use positive controls, such as siRNA-transfected cells that have been scrambled.
Response: The control siRNA we used is in fact a scrambled siRNA as indicated in chapter 2.3 of the Methods section.
- Provide a thorough explanation of the data and conduct out in vivo experiments to comprehend TAp73's function in PDAC's EMT regulation.
Response: The in vivo experiments in TAp73 genetically engineered mouse model with measurement of EMT regulation have already been performed by Thakur and colleagues (see Ref. 13). We should point out that our study was designed as an in vitro study. To perform animal experiments with xenotransplanted human cells we are unlikely to obtain a permitting vote from our ethical committee as long as our corresponding in vitro data have not been published.
8- The Authors also need to perform the invasion assay.
Response: We have now performed invasion assays with PANC-1 cells. In the revised version these data are shown in Supplementary materials as new Figure S4. Technical details have been added to the Methods section 2.6.
Reviewer 2 Report
In the manuscript, entitled, “ TAp73 inhibits EMT and cell migration in pancreatic cancer cells through promoting SMAD4 expression and SMAD4 dependent inhibition of ERK activation”, the authors aim to show the role of TAp73 in promoting the canonical SMAD-dependent TGF-b signaling and inhibiting the non-canonical ERK signaling. They further show how this modulation affects cell migration in human PDAC cell lines.
1. The main concern of this manuscript is its novelty. The authors themselves keep citing Thakur et al 2016 paper in Cell Death and Differentiation. Initially, I thought that may be Thakur et al did not test any of their findings in human PDAC cell lines. But carefully looking at that manuscript, showed me that they did use PANC1 (Fig. 3, 4 and 6) - the same cell line used by the authors of this manuscript. It is not clear why the authors want to publish very similar findings to a paper that was published in 2016.
2. On Page 6, lines 244-246, the authors are talking about “the inhibitory effect of TGF-b1 treatment on CDH1 was reduced from 80% to 30%”. But the graph in Fig. 2B doesn't show this. It rather shows that inhibitory effect of TGF-b1 on CDH1 (in the graph denoted as ECAD) is completely lost in TAp73 deficient situation and it rather gets upregulated in these cells on TGF-b1 treatment.
3. In simple summary (lines 17-18), the authors mention, “TAp73 promotes epithelial mesenchymal transition by inducing the expression of epithelial markers while suppressing that of mesenchymal markers. If this is the case, then TAp73 suppresses the transition of epithelial cells to mesenchymal phenotype. Replace “promotes” epithelial mesenchymal transition with “suppresses” epithelial mesenchymal transition.
4. On Page 4, line 176 there are two “in”. Delete repeated word.
5. In all the graphs showing mRNA levels, change the y-axis title to “Relative mRNA expression”.
Quality of english is ok. Try to avoid very long sentences, makes the text confusing. Double check the text for typos and repeated words.
Author Response
Dear Editor:
We thank the reviewers for their critical and generally very positive comments on our manuscript. However, we only disagree with some notions of Reviewer 2 as detailed below. We believe that incorporating the suggestions and recommendations of all three reviewers into the revised version has significantly enhanced the quality of our manuscript.
In the manuscript, entitled, “ TAp73 inhibits EMT and cell migration in pancreatic cancer cells through promoting SMAD4 expression and SMAD4 dependent inhibition of ERK activation”, the authors aim to show the role of TAp73 in promoting the canonical SMAD-dependent TGF-b signaling and inhibiting the non-canonical ERK signaling. They further show how this modulation affects cell migration in human PDAC cell lines.
- The main concern of this manuscript is its novelty. The authors themselves keep citing Thakur et al 2016 paper in Cell Death and Differentiation. Initially, I thought that may be Thakur et al did not test any of their findings in human PDAC cell lines. But carefully looking at that manuscript, showed me that they did use PANC1 (Fig. 3, 4 and 6) - the same cell line used by the authors of this manuscript. It is not clear why the authors want to publish very similar findings to a paper that was published in 2016.
Response: We believe that using the same cell line (PANC-1) as in the Thakur study from 2016 cannot be regarded a problem by itself. On the contrary, we have chosen PANC-1 cells intentionally to validate our migration data with those of Thakur et al. It must be pointed out that our migration data with PANC-1 cells are completely different from those in the Thakur study. Our focus in measuring migration was the response of TAp73 knockdown cells to TGF-β1, which was not analysed at all in the Thakur study. However, in order to show the response to TGF-β of the TAp73 knockdown cells, it was mandatory to also measure and display in the graphs non-TGF-β1 treated PANC-1 cells as control.
It should also be mentioned that even our migration data with the non-TGF-β1 treated control PANC-1 cells were not identical with those in the Thakur study for two reasons: i) we employed another assay type/method (xCELLigence real-time measurement). This technique provides kinetic information over several hours (with 4 data points per hour), while the migration assay of Thakur was a conventional Boyden chamber and hence endpoint assay providing only 2 data points, one for time zero and the second one at assay termination ii) we performed the xCELLigence assay in a chemokinesis setup measuring basal/spontaneous migratory activity, while Thakur et al used a chemotactic setup measuring directed migratory activity against an FBS gradient. We have now mentioned these differences in the Discussion section (5th paragraph) of the revised version. In addition, in contrast to our ms., Thakur and coworkers did neither provide migration data from a second human PDAC cell line nor did they provide data on ectopic expression of the alpha and beta isoforms of TAp73. This means that they were unable to draw conclusions on the isoform responsible for inhibiting PANC-1 migratory/invasive activity as their p73 siRNA was likely to inhibit both isoforms. Last but not least and in contrast to the Thakur study, we have functionally analysed – using RNA- and dominant-negative interference - the role of SMAD4 as a mediator of TAp73a driven inhibition of cell migration.
The same situation as with the migration assay actually applies to the invasion assay that we have included in the revised version (Figure S4) in response to a request from Reviewer 1.
Reviewer 2 also mentioned PANC-1 data in Figs. 4 and 6 of the Thakur study. These data are not relevant for our ms. as they show measurement of biglycan (a factor not studied in our ms., Fig. 4), or analyse migratory activity in response to kinase inhibitors (Fig. 6) whose substrates (p38 MAPK and PI3K/AKT) were not studied by us either. Neither of the two kinase inhibitors used by Thakur et al. does inhibit ERK activation, which means that there is no overlap with respect to ERK activation data between our ms. and that of Thakur et al.
We sincerely hope that we have convincingly pointed out here why we feel that the statements "lack of novelty" and "publishing of very similar findings" are not justified. We regret, however, that we did not mention in our ms. that Thakur et al have already measured basal migratory and invasive activities in PANC-1 TAp73 knockdown cells albeit using a different assay type and experimental conditions. This has now been rectified by adding this piece of information to the Discussion section (5th and 6th paragraph).
- On Page 6, lines 244-246, the authors are talking about “the inhibitory effect of TGF-b1 treatment on CDH1 was reduced from 80% to 30%”. But the graph in Fig. 2B doesn't show this. It rather shows that inhibitory effect of TGF-b1 on CDH1(in the graph denoted as ECAD) is completely lost in TAp73 deficient situation and it rather gets upregulated in these cells on TGF-b1 treatment.
Response: This was in fact a misunderstanding. We have referred in our description to the very left bar (control siRNA, untreated cells), while the reviewer referred to the third bar (p73 siRNA, untreated cells). This confusion has been removed by rephrasing the description.
- In simple summary (lines 17-18), the authors mention, “TAp73 promotes epithelial mesenchymal transition by inducing the expression of epithelial markers while suppressing that of mesenchymal markers. If this is the case, then TAp73 suppressesthe transition of epithelial cells to mesenchymal phenotype. Replace “promotes” epithelial mesenchymal transition with “suppresses” epithelial mesenchymal transition.
Response: This is of course true. We apologize for this error and have removed it.
- On Page 4, line 176 there are two “in”. Delete repeated word.
Response: Done
- In all the graphs showing mRNA levels, change the y-axis title to “Relative mRNA expression”.
Response: Done
Try to avoid very long sentences, makes the text confusing. Double check the text for typos and repeated words.
Response: Done
Reviewer 3 Report
The manuscript investigates the role of TAp73, a p53 homolog, in pancreatic ductal adenocarcinoma (PDAC) and its influence on tumor development and progression. The study utilized two human PDAC cell lines to explore the function of TAp73. The findings revealed that TAp73 upregulated the expression of epithelial markers (ECAD and SMAD4) while downregulating the expression of the mesenchymal marker SNAIL in human PDAC cells. TAp73 also facilitated TGF-β signaling through a SMAD4-dependent pathway, regulating the expression of TGF-β/SMAD target genes involved in epithelial-mesenchymal transition (EMT). Additionally, TAp73 inhibited the constitutive activation of the non-Smad signaling pathway ERK, thereby suppressing cell migration and invasion. The study demonstrated that SMAD4 mediated the effects of TAp73 on EMT, cell motility, and signaling pathways. Loss of TAp73 resulted in reduced SMAD4 expression and increased ERK activation, leading to enhanced EMT and migration/invasion. These findings highlight the crucial role of TAp73 in inhibiting EMT, regulating TGF-β signaling, and suppressing ERK-mediated processes in PDAC. The authors propose that TAp73 could have implications for therapeutic strategies targeting TGF-β in PDAC. Understanding the role of TAp73 and its impact on TGF-β signaling could aid in patient selection for TGF-β inhibitors and improve treatment outcomes. Furthermore, the measurement of TAp73, SMAD4, and pERK levels could serve as predictive biomarkers to determine the preferential activation of oncogenic or tumor suppressive pathways in individual patients. Overall, the study provides insights into the complex involvement of TGF-β and TAp73 in pancreatic tumorigenesis and highlights the therapeutic potential of targeting TGF-β signaling in PDAC treatment.
Improvement:
Functional assays: It would be valuable to include functional assays to assess the effects of manipulating TAp73 or TGF-β signaling on tumor growth, metastasis, and treatment response. This would provide a more comprehensive understanding of the biological consequences and potential therapeutic benefits. Some suggested assays include:
1) Metastasis assays: Investigate the role of TAp73 and TGF-β signaling in metastasis by assessing PDAC cells' ability to invade, migrate, and form metastatic colonies.
2) Treatment response assays: Evaluate the impact of manipulating TAp73 or TGF-β signaling on the response of PDAC cells to different treatment modalities, such as chemotherapy or targeted therapy.
3) Combination therapy assays: Investigate the potential synergistic effects of targeting TAp73 or TGF-β signaling in combination with existing PDAC therapies.
Minor:
1) Line 240, ‘pf’, should be ‘of’.
Author Response
Dear Editor:
We thank the reviewers for their critical and generally very positive comments on our manuscript. However, we only disagree with some notions of Reviewer 2. We believe that incorporating the suggestions and recommendations of all three reviewers into the revised version has significantly enhanced the quality of our manuscript.
Reviewer 3
The manuscript investigates the role of TAp73, a p53 homolog, in pancreatic ductal adenocarcinoma (PDAC) and its influence on tumor development and progression. The study utilized two human PDAC cell lines to explore the function of TAp73. The findings revealed that TAp73 upregulated the expression of epithelial markers (ECAD and SMAD4) while downregulating the expression of the mesenchymal marker SNAIL in human PDAC cells. TAp73 also facilitated TGF-β signaling through a SMAD4-dependent pathway, regulating the expression of TGF-β/SMAD target genes involved in epithelial-mesenchymal transition (EMT). Additionally, TAp73 inhibited the constitutive activation of the non-Smad signaling pathway ERK, thereby suppressing cell migration and invasion. The study demonstrated that SMAD4 mediated the effects of TAp73 on EMT, cell motility, and signaling pathways. Loss of TAp73 resulted in reduced SMAD4 expression and increased ERK activation, leading to enhanced EMT and migration/invasion. These findings highlight the crucial role of TAp73 in inhibiting EMT, regulating TGF-β signaling, and suppressing ERK-mediated processes in PDAC. The authors propose that TAp73 could have implications for therapeutic strategies targeting TGF-β in PDAC. Understanding the role of TAp73 and its impact on TGF-β signaling could aid in patient selection for TGF-β inhibitors and improve treatment outcomes. Furthermore, the measurement of TAp73, SMAD4, and pERK levels could serve as predictive biomarkers to determine the preferential activation of oncogenic or tumor suppressive pathways in individual patients. Overall, the study provides insights into the complex involvement of TGF-β and TAp73 in pancreatic tumorigenesis and highlights the therapeutic potential of targeting TGF-β signaling in PDAC treatment.
Improvement:
Functional assays: It would be valuable to include functional assays to assess the effects of manipulating TAp73 or TGF-β signaling on tumor growth, metastasis, and treatment response. This would provide a more comprehensive understanding of the biological consequences and potential therapeutic benefits. Some suggested assays include:
- Metastasis assays: Investigate the role of TAp73 and TGF-β signaling in metastasis by assessing PDAC cells' ability to invade, migrate, and form metastatic colonies.
Response: We appreciate the reviewers’ suggestion, however, this would require animal experiments, for which it will be difficult to obtain a permitting vote from the local ethical committee prior to publication of the in vitro data. We should point out that this study was designed as an in vitro study to generate these data. Moreover, metastasis has been analysed already by Thakur and colleagues in their TAp73-deficient mouse model (see Ref. 13, Figure 3h). We are nevertheless planning these experiments and have added this to the last paragraph of the Discussion section.
2) Treatment response assays: Evaluate the impact of manipulating TAp73 or TGF-β signaling on the response of PDAC cells to different treatment modalities, such as chemotherapy or targeted therapy.
Response: The focus of our study was clearly on the impact of TAp73 and SMAD4 on ERK activation and cell migration/invasion. We believe that including other cellular responses such as response to therapy would exceed the scope of our ms. Definitely, analysing the role of TAp73 here would be novel and exciting, however, this is scheduled to be part of a separate study currently performed by us. The outcome of manipulating TGF-β signaling on the response of PDAC cells to different treatment modalities, such as chemotherapy or targeted therapy has already been investigated in a couple of previous studies.
- Combination therapy assays: Investigate the potential synergistic effects of targeting TAp73 or TGF-β signaling in combination with existing PDAC therapies.
Response: This is a very good suggestion. However, please see my response to 2).
Minor:
- Line 240, ‘pf’, should be ‘of’.
Response: Done
Round 2
Reviewer 1 Report
The Authors have significantly improved the paper and performed all required experiments as suggested.
This section looks good
Reviewer 2 Report
Authors have satisfied my concerns.